# Emergence of the Ug99 lineage of the wheat stem rust pathogen through somatic hybridisation

Feng Li [1,10], Narayana M. Upadhyaya [2,10], Jana Sperschneider[3], Oadi Matny [1], Hoa Nguyen-Phuc [1], Rohit Mago[2], Castle Raley[4,8], Marisa E. Miller[1,9], Kevin A.T. Silverstein [5], Eva Henningsen [1], Cory D. Hirsch [1], Botma Visser[6], Zacharias A. Pretorius[6], Brian J. Steffenson [1], Benjamin Schwessinger[7], Peter N. Dodds [2]* & Melania Figueroa [2]*

Parasexuality contributes to diversity and adaptive evolution of haploid (monokaryotic) fungi. However, non-sexual genetic exchange mechanisms are not defined in dikaryotic fungi (containing two distinct haploid nuclei). Newly emerged strains of the wheat stem rust pathogen, *Puccinia graminis* f. sp. *tritici* (*Pgt*), such as Ug99, are a major threat to global food security. Here, we provide genomics-based evidence supporting that Ug99 arose by somatic hybridisation and nuclear exchange between dikaryons. Fully haplotype-resolved genome assembly and DNA proximity analysis reveal that Ug99 shares one haploid nucleus genotype with a much older African lineage of *Pgt*, with no recombination or chromosome reassortment. These findings indicate that nuclear exchange between dikaryotes can generate genetic diversity and facilitate the emergence of new lineages in asexual fungal populations.

[1] Department of Plant Pathology, University of Minnesota, St. Paul, MN 55108, USA. [2] Commonwealth Scientific and Industrial Research Organisation, Agriculture and Food, Canberra ACT 2601, Australia. [3] Biological Data Science Institute, The Australian National University, Canberra ACT 2601, Australia. [4] Leidos Biomedical Research, Frederick, MD 21702, USA. [5] Minnesota Supercomputing Institute, Minneapolis, MN 55455, USA. [6] University of the Free State, Bloemfontein 9301 Free State, South Africa. [7] Research School of Biology, The Australian National University, Canberra ACT 2601, Australia. [8] Present address: The George Washington University, Washington, DC 20052, USA. [9] Present address: Pairwise, Durham, NC 27709, USA. [10] These authors contributed equally: Feng Li, Narayana M. Upadhyaya *email: peter.dodds@csiro.au; melania.figueroa@csiro.au

Generation of genetic diversity is crucial for the evolution of new traits, with mutation and sexual recombination as the main drivers of diversity in most eukaryotes. However, many species in the fungal kingdom can propagate asexually for extended periods and therefore understanding alternative mechanisms contributing to genetic diversity in asexual populations has been of great interest[1,2]. Some fungi can use a parasexual mechanism to exchange genetic material independently of meiosis[2]. This process involves anastomosis of haploid hyphae and fusion of two nuclei to generate a single diploid nucleus, which subsequently undergoes progressive chromosome loss to generate recombinant haploid offspring. Parasexuality has been described in members of the ascomycete phylum (64% of described fungal species) in which the dominant asexually propagating form is haploid[3]. However, in basidiomycete fungi (34% of described species), the predominant life stage is generally dikaryotic, with two different haploid nuclei maintained within each individual[3]. The role of non-sexual genetic exchange between such dikaryons in generating genetic diversity is not known.

Basidiomycetes include many fungi with critical ecosystem functions, such as wood decay and plant symbiosis, as well as agents of important human and plant diseases[1]. Rust fungi (subphylum Pucciniomycotina) comprise over 8000 species including many pathogens of major agricultural and ecological significance[4]. These organisms are obligate parasites with complex life cycles that can include indefinite asexual reproduction through infectious dikaryotic urediniospores. Early researchers speculated that rust fungi can exchange genetic material during the asexual phase based on the isolation of new strains, after co-infection with two potential parental isolates, with novel virulence phenotypes[5–8]. However, these hypotheses could not be tested molecularly at the time. Some naturally occurring rust pathotypes have been suggested to have arisen by somatic hybridisation and genetic exchange based on limited molecular evidence of shared isozyme or random amplified polymorphic DNA (RAPD) markers[9,10]. Mechanisms underlying genetic exchange are unknown, but may involve hyphal anastomosis followed by nuclear exchange and/or nuclear fusion and recombination[11]. Recent advances in assembling complete karyon sequences in rust fungi[12,13] provide the opportunity to definitively detect and discriminate between nuclear exchange and recombination.

The Ug99 strain (race TTKSK) of the wheat stem rust pathogen Puccinia graminis f. sp. tritici (Pgt) presents a significant threat to global wheat production[14]. It was first detected in Uganda in 1998 and described in 1999[15], and has since given rise to an asexual lineage that has spread through Africa and the Middle East causing devastating epidemics[14]. The origin of the Ug99 lineage is unknown, although it is genetically distinct from other Pgt races[16,17]. This indicates that Ug99 is not likely derived by mutation of longstanding stem rust asexual lineages such as the race 21 group, which has been predominant in southern Africa at least since the 1920's and spread to Australia in the 1950's[18–20].

Here, we generate haplotype-phased genome references for the original Ug99 isolate collected in Uganda[15] and an Australian Pgt isolate of pathotype 21-0[20,21]. We show by genome comparison that Ug99 shares one haploid nucleus genotype with Pgt21-0, with no recombination or chromosome reassortment. This indicates that Ug99 arose by somatic hybridisation and nuclear exchange between an African member of the Pgt 21 lineage and an unknown isolate. Thus, nuclear exchanges between dikaryotic fungi can contribute to the emergence of new variants with significant epidemiological impacts.

## Results

**Haplotype-phased genome assembly.** A single pustule derived from the original Ug99 isolate of Pgt[15] was purified and its pathotype was confirmed using the standard wheat differential set (Supplementary Data 1). We generated polished long-read genome assemblies for both Ug99 and the Australian stem rust isolate Pgt21-0[18] using single-molecule real-time (SMRT) and Illumina sequence data (Supplementary Tables 1 and 2). Genome assembly of Pgt21-0 resulted in 410 contigs with a total size of 177 Mbp and a contig $N_{50}$ of 1.26 Mbp (Supplementary Table 3). Similarly, the size of the genome assembly of Ug99 was 176 Mbp represented in 514 contigs with contig $N_{50}$ of 0.97 Mbp. Thus, both assemblies were twice the size of a collapsed haploid assembly previously generated for a North American Pgt isolate (88 Mbp)[22], suggesting that in each assembly the sequences of the two haploid karyons were fully represented independently. Consistent with this, both genome assemblies contained over 96% of conserved fungal BUSCO genes, with the majority present in two copies (Supplementary Table 3). Furthermore, the Pgt21-0 assembly contained 69 telomeres, out of a total of 72 expected for a dikaryotic genome given the known haploid chromosome number of eighteen[23]. To identify sequences representing alternate haplotypes within each assembly we developed a gene synteny approach to assign contigs to groups representing paired homologous haplotypes (Fig. 1). Using this approach, homologous pairs of sequences from each haplotype were assigned to 44 bins in Pgt21-0 and 62 bins in Ug99, which represented over 94% of each assembly (Supplementary Table 3 and Supplementary Data 2). Three of the 18 chromosomes in Pgt21-0, and two in Ug99 seemed to be fully assembled as these bins contained telomere sequences at each end.

The AvrSr50 and AvrSr35 genes encode dominant avirulence factors recognised by wheat resistance genes[23,24]. These two genes are located in close proximity (<15 kbp) to each other in the genome assemblies and both haplotypes of this locus were assembled as alternate contigs in Pgt21-0 and Ug99 (Fig. 2a). Both isolates were heterozygous for AvrSr50 with one allele containing a ~26-kbp-insertion. Pgt21-0 was also heterozygous for AvrSr35, with one allele containing a 400-bp-MITE insertion previously described[24]. PCR amplification from the Ug99 strain had previously identified only a single AvrSr35 allele, suggesting that it was homozygous in Ug99[24]. However, the Ug99 genome assembly contained a second allele of this gene, which contained a 57-kbp-insertion that would have prevented its PCR amplification in the Salcedo et al. study[24]. The presence of this insertion in Ug99 was supported by DNA read (PacBio and Illumina) alignments across this genomic region and confirmed by DNA amplification and amplicon sequencing of flanking border regions (Supplementary Fig. 1). Thus, Ug99 is also heterozygous for avirulence on Sr35, and may therefore mutate to virulence on this wheat resistance gene more readily than if it were homozygous. This is an important finding as it will inform Sr35 deployment strategies against Ug99. Strikingly, the AvrSr35-virSr50 haplotype of this locus is very similar in structure in Ug99 and Pgt21-0 (Fig. 2a) and shares >99% sequence identity, while the two alternate haplotypes are quite different. We therefore compared the larger genomic regions containing these loci in each isolate: namely bin 06 in Pgt21-0, which is ~3.5 Mbp and includes telomeres at both ends, and bins 15 (1.8 Mbp) and 23 (1.2 Mbp) in Ug99 (Supplementary Fig. 2a). One haplotype (designated A) was >99.7% identical in Ug99 and Pgt21-0, while the other two haplotypes (B and C) were highly divergent from each other and from haplotype A (Fig. 2b, Supplementary Fig. 2b and Supplementary Table 4). Only 71–78% of the sequences from each haplotype aligned, with an average identity of ~95% across the aligned regions, yielding total identities of only 68–76%

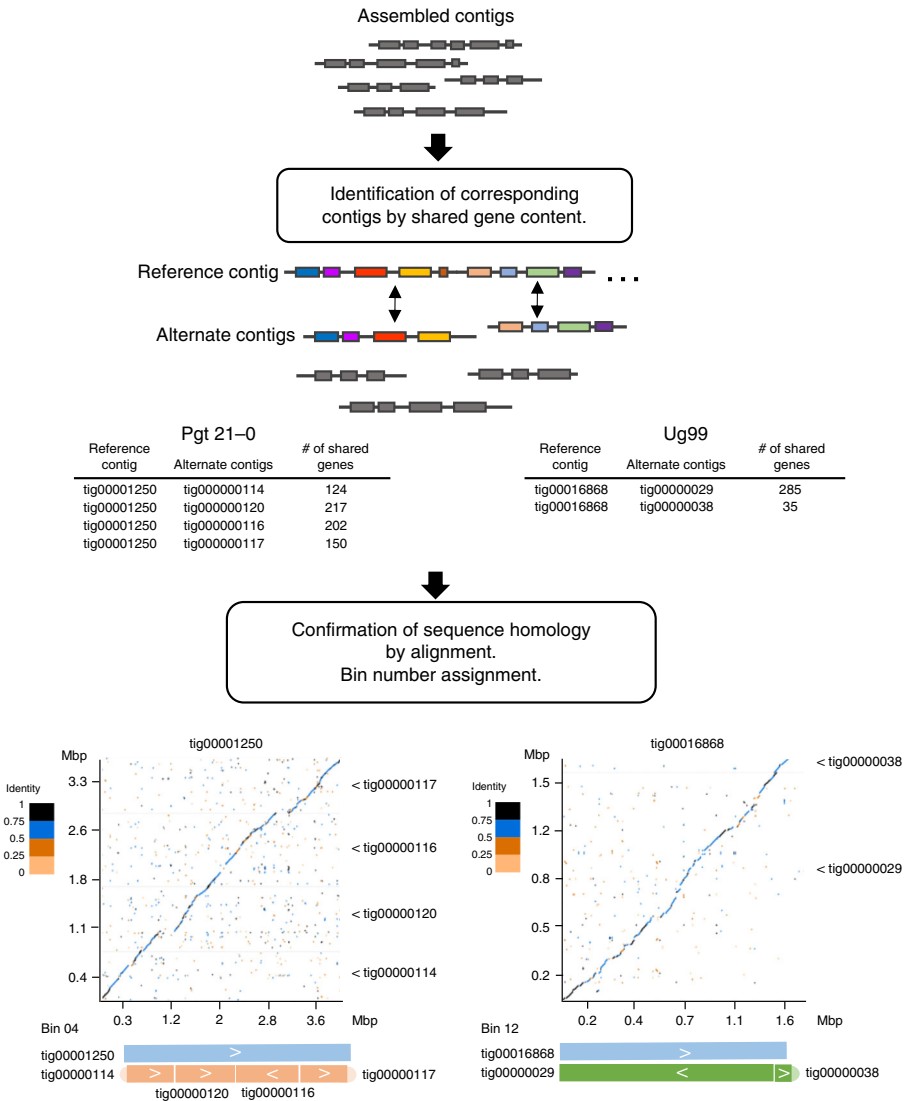

**Fig. 1** Strategy to identify homologous contigs in genome assemblies by gene synteny. To detect shared content, *Pgt* gene models[21] (grey and coloured boxes) were aligned to the genome assemblies and the contig positions of the top two hits of each gene were recorded. Contigs containing at least five shared genes were considered as potential haplotype pairs. Sequence collinearity between contigs was assessed by alignment, and homologous matching contigs were assigned to bins. Examples shown are for Bin04 and Bin12 from *Pgt*21-0 and Ug99, respectively

between these three distinct haplotypes. The high similarity between the A haplotypes of this chromosome suggested that Ug99 and *Pgt*21-0 may share large portions of their genomes, potentially up to an entire haploid genome copy.

**Whole-genome haplotype assignment and comparison**. We used a read subtraction and mapping approach (Fig. 3a, b) to identify genome regions in the Ug99 and *Pgt*21-0 assemblies that showed high similarity and may be derived from a shared haplotype. Illumina reads from each isolate were mapped to the genome reference of the other isolate. Reads that failed to map were retained, and this subtracted read set, in which sequences common to both isolates had been removed, was re-mapped to their original genome reference. The subtracted read coverage depth was compared to the coverage depth obtained when using all reads (Supplementary Fig. 3a, b). Contigs representing sequences shared by both isolates (designated as haplotype A) displayed a very low subtracted read coverage depth (Supplementary Data 3). In contrast, sequences unique to *Pgt*21-0 or Ug99 (designated as haplotypes B or C, respectively) retained a

relatively high subtracted read coverage depth. Some contigs in each assembly appeared to be chimeric with distinct regions assigned to opposite haplotypes, and these contigs were divided into separate fragments (Supplementary Data 4) for subsequent haplotype comparisons. Approximately half of each genome assembly was assigned to either the A, B or C haplotypes (Fig. 3c) and importantly one set of homologous sequences from each bin was assigned to each haplotype (Supplementary Data 3). The A, B and C haplotype sets contained 95–96% of conserved fungal BUSCO genes (Fig. 3d), indicating that each represents a full haploid genome equivalent. Consistent with this, the haplotypes were highly contiguous (Fig. 4). Overall sequence identity between the A haplotypes of *Pgt*21-0 and Ug99 was over 99.5%, with structural variation (large insertions/deletions) representing only 0.46% of the haplotype sizes, and with just 0.08% sequence divergence in aligned regions (Fig. 4a, Table 1, and Supplementary Table 5). In contrast, only 91–93% of the A, B or C haplotypes could be aligned to each other, with an average sequence identity of ~96% across the aligned regions, giving total identities of 87–91% (Fig. 4b–d, Table 1 and Supplementary Table 5).

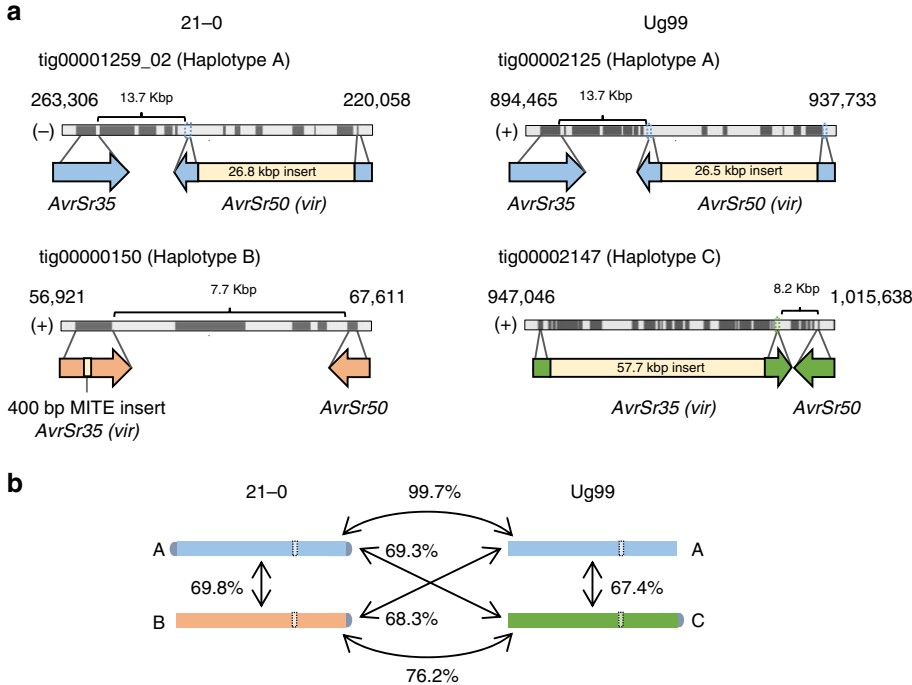

**Fig. 2** A common haplotype containing *AvrSr50* and *AvrSr35* is shared between *Pgt*21-0 and Ug99. **a** Diagram of genomic regions containing *AvrSr50* and *AvrSr35* alleles in *Pgt*21-0 and Ug99. Numbers above tracks correspond to contig coordinates and the sense of the DNA strand is indicated as + or −. Predicted gene models (including introns) are depicted as dark grey boxes and intergenic spaces are shown in light grey. Coloured arrows indicate location and direction of *AvrSr50* and *AvrSr35* genes, with size and position of insertions shown in yellow. Intergenic distances between *AvrSr50* and *AvrSr35* are indicated by brackets. **b** Total sequence identity between contigs representing homologous chromosomes of different haplotypes (coloured bars) containing the *AvrSr50/AvrSr35* locus (dotted white boxes). Telomere sequences are represented in grey. Chromosome size = ~3.5 Mbp

Structural variation between these haplotypes accounted for 6.7–8.6% of the haploid genome sizes. There were only ~9000 SNPs (0.1/kbp) between the two A haplotypes, versus 876,000 to 1.4 million SNPs (11–18/kbp) between the A, B and C pairs, which is consistent with estimates of heterozygosity levels in *Pgt*21-0 (haplotypes A and B) based on variant detection from Illumina read mapping[21]. The high similarity between the A haplotypes, and divergence between A, B and C haplotypes was also supported by Illumina read coverage and SNP calling analysis (Supplementary Fig. 3c–e) showing that Ug99 and *Pgt*21-0 share one nearly identical haploid genome copy.

**Assessment of inter-nuclear recombination**. We tested two hypotheses that could explain the shared haplotype between Ug99 and *Pgt*21-0: (1) Ug99 arose by a somatic hybridisation event in which an isolate of the race 21 lineage donated an intact nucleus of the A haplotype (Fig. 5a); and (2) Ug99 arose by a sexual cross in which one haploid pycnial parent was derived from a race 21 lineage isolate after meiosis (Fig. 5b). Under both scenarios, the A haplotype of Ug99 represents one entire haploid nucleus that was derived from the race 21 lineage isolate. In the nuclear exchange scenario, the *Pgt*21-0 A haplotype represents a single nucleus donated intact to generate Ug99. However, under the sexual cross model, this *Pgt*21-0 haplotype would include segments of both nuclear genomes that were combined by crossing over and chromosome reassortment after karyogamy and meiosis. Although recombination frequency has not been measured in *Pgt*, an average of 115 recombination events per haploid genome was detected during meiosis in the related flax rust fungus (*Melampsora lini*), which also has 18 chromosomes[25]. Because the *Pgt*21-0 and Ug99 genome assemblies represent the phased dikaryotic state of each isolate, all correctly phased contigs in Ug99 should be either A or C haplotype, while

those in *Pgt*21-0 would include mixed haplotype contigs only if the sexual cross hypothesis is correct. In fact, just 19 contigs in the Ug99 assembly contained adjacent regions of either the A or C haplotype. These appeared to result from haplotype phase swap artefacts during genome assembly, since all of the junctions occurred between phase blocks (i.e. opposite gaps between the corresponding alternate contigs). Furthermore, examination of Illumina read mapping to these regions revealed that these sites contained either collapsed haplotype sequences, non-unique sequences or discontinuities in read coverage (Supplementary Fig. 4), indicative of assembly errors disrupting phase information across the junction. Likewise, 31 contigs of mixed haplotype in the *Pgt*21-0 assembly all contained likely phase swap artefacts (Supplementary Fig. 4). To experimentally distinguish between phase-swap assembly artefacts and meiotic recombination events, we used Hi-C chromatin cross-linking proximity analysis[26] to assess physical linkage between contigs in the *Pgt*21-0 assembly. About 90% of all read pair connections were between contigs of the same haplotype. For each of the chimeric contigs, the separated A and B fragments showed significantly more connections to contigs of the same haplotype than to contigs of the other haplotype, including other fragments of the original chimeric contig (Supplementary Data 5). These observed physical linkages confirmed that all of the mixed haplotype contigs in *Pgt*21-0 resulted from phase swap errors during the genome assembly process, and do not correspond to sites of genetic recombination that could have occurred during meiosis under the sexual origin hypothesis. Furthermore, only four structural variants larger than 10 kbp were detected between the A haplotypes of Ug99 and *Pgt*21-0. Three of these were deletions in Ug99, while the fourth was a tandem repeat duplication, indicating that there was no novel genetic content in Ug99 haplotype A compared to *Pgt*21-0.

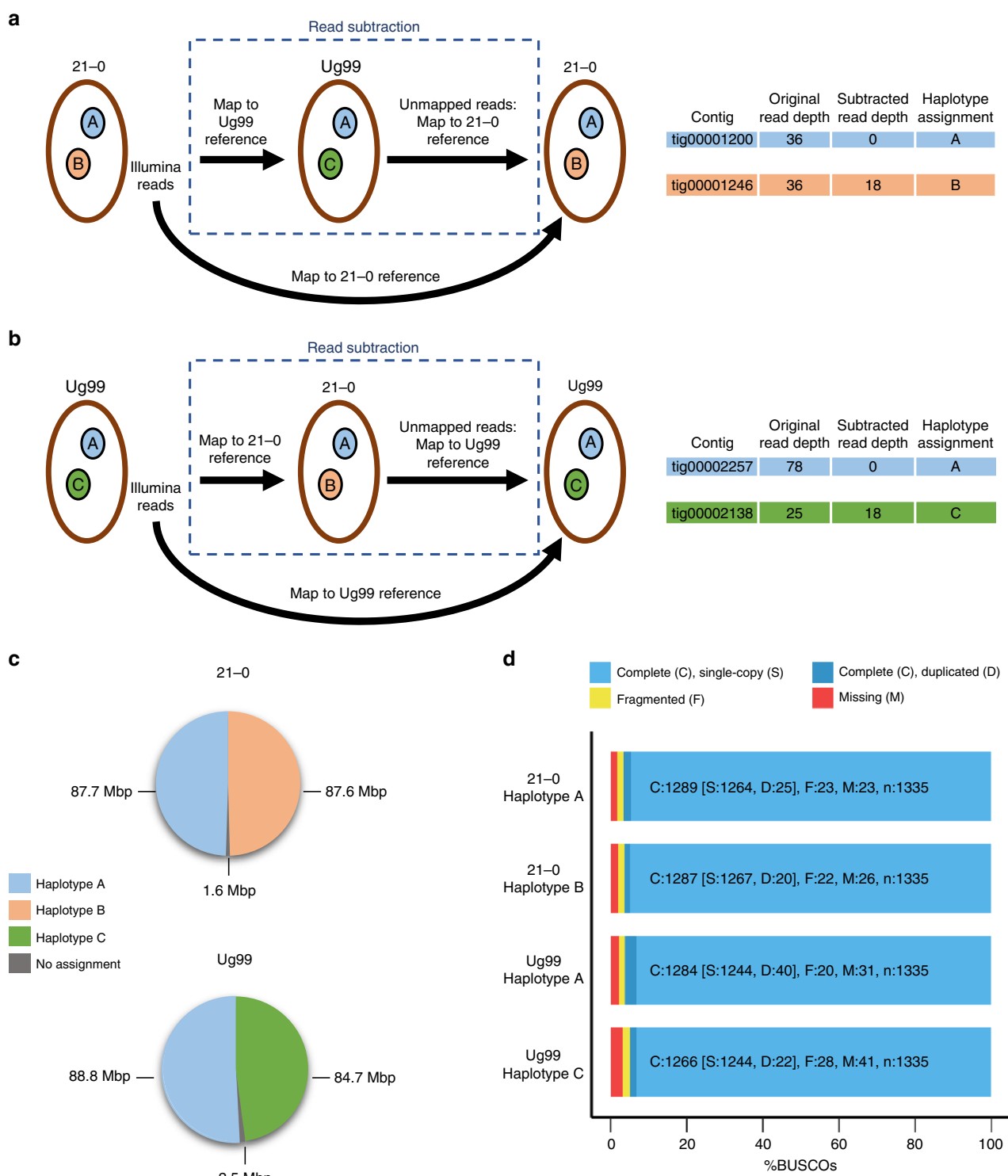

**Fig. 3** Haplotype assignment by read subtraction and mapping process. **a** Illumina reads from *Pgt*21-0 were mapped to the Ug99 genome assembly at high stringency. Unmapped reads derived from divergent regions of the B haplotype were retained and then mapped to the *Pgt*21-0 genome assembly. Read coverage of individual contigs with the original and subtracted reads were compared to designate haplotypes as either A or B. **b** The same process was followed with reads from Ug99 subtracted against the *Pgt*21-0 reference to designate the A and C haplotypes. **c** Pie chart showing proportion and total sizes of contigs assigned to haplotypes A, B or C or unassigned in *Pgt*21-0 and Ug99 assemblies. **d** BUSCO analysis to assess completeness of haplotype genome assemblies. Bars represent the percentage of total BUSCOs as depicted by the colour key

**Chromosome assembly and assessment of reassortment.** Combining Hi-C scaffolding data with the bin and haplotype assignment information for the *Pgt*21-0 assembly allowed us to construct 18 chromosome pseudomolecules for each of the A and

B haplotypes (Fig. 6a, Supplementary Table 6 and Supplementary Data 6). These covered a total of 170 Mbp and ranged from 2.8 to 7.3 Mbp in size, consistent with relative chromosome sizes from karyotype analysis[27]. The A and B chromosomes were collinear

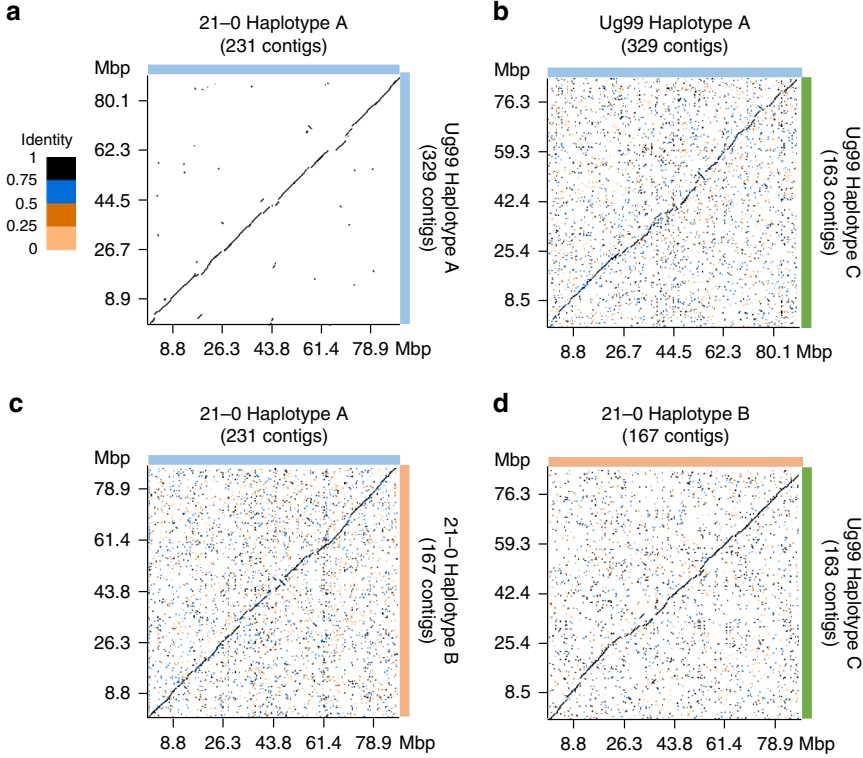

**Fig. 4** *Pgt*21-0 and Ug99 share one nearly identical haploid genome. **a–d** Dot plots illustrating sequence alignment of complete haplotypes. *X*- and *y*-axes show cumulative size of the haplotype assemblies depicted by coloured bars to the right and top of the graphs. Colour key indicates sequence identity ratios for all dot plots

| **Table 1 Intra- and inter-isolate sequence comparison of entire haplotypes in Ug99 and *Pgt*21-0** | | | | | |
| --- | --- | --- | --- | --- | --- |
| | **Sequence similarity** | | **Structural variation** | | |
| | | | | **Total variant size** | |
| **Isolate comparison** | **Bases aligned (%)** | **Sequence divergence (%)** | **Number of variants** | **Mbp** | **% of genome** |
| 21-0A vs Ug99 A | 99.64 | 0.08 | 491 | 0.82 | 0.46 |
| Ug99 A vs Ug99 C | 91.52 | 4.08 | 2571 | 13.69 | 7.88 |
| 21-0A vs *Pgt*21-0 B | 91.38 | 4.19 | 2696 | 15.01 | 8.56 |
| 21-0 B vs Ug99 C | 93.44 | 2.4 | 1910 | 11.50 | 6.69 |

except for two translocation events (Fig. 6b). In each case these were supported by contigs that spanned the translocation breakpoints. Re-scaffolding the separated fragments of these contigs using Hi-C data supported the original contig assembly, indicating that these are true translocation events within the A or B genomes. The haplotype A chromosomes showed high collinearity with the Ug99 A haplotype contigs (Fig. 6c).

Approximately 65% of the total Hi-C read pairs represented links between physically contiguous sequences on the same chromosome, while the remaining pairs connected sites distributed across the genome. Because Hi-C DNA crosslinking is performed in intact cells, these non-scaffolding linkages should preferentially form between chromosomes that are located in the same nucleus. Indeed, all chromosomes of the A haplotype showed a much higher proportion of Hi-C read pair links to other chromosomes of the A haplotype (~85%) than to chromosomes of the B haplotype (~15%) (Fig. 6d), indicating that they are all located in the same nucleus. Thus, the A haplotype of Ug99 derives from a single nucleus of *Pgt*21-0. Similarly, 17 of the B haplotype chromosomes in *Pgt*21-0 showed stronger linkage to other B chromosomes (~90%) than to A chromosomes (~10%)

(Fig. 6e). However, chromosome 11B showed the inverse, suggesting that both homologues of this chromosome are in the same nucleus. This implies that a single chromosome exchange event occurred during asexual propagation of the *Pgt*21-0 isolate, after its divergence from the race 21 lineage branch giving rise to Ug99.

Overall the whole-genome comparison data demonstrate that Ug99 shares one full haploid nuclear genome with the *Pgt*21-0 isolate with no recombination events within chromosomes and no reassortment of chromosomes from different nuclei. These facts are inconsistent with a sexual origin, and strongly support that the Ug99 lineage arose by a somatic hybridisation event involving one parent derived from the African race 21 lineage and another parent of unknown origin exchanging whole nuclei (Fig. 7).

**Comparison of gene content between haplotypes**. Annotation of the *Pgt*21-0 and Ug99 genome assemblies predicted similar gene numbers (~18,000) in each haplotype (Supplementary Table 7). Gene orthology analysis indicated that 65–70% of genes in each of

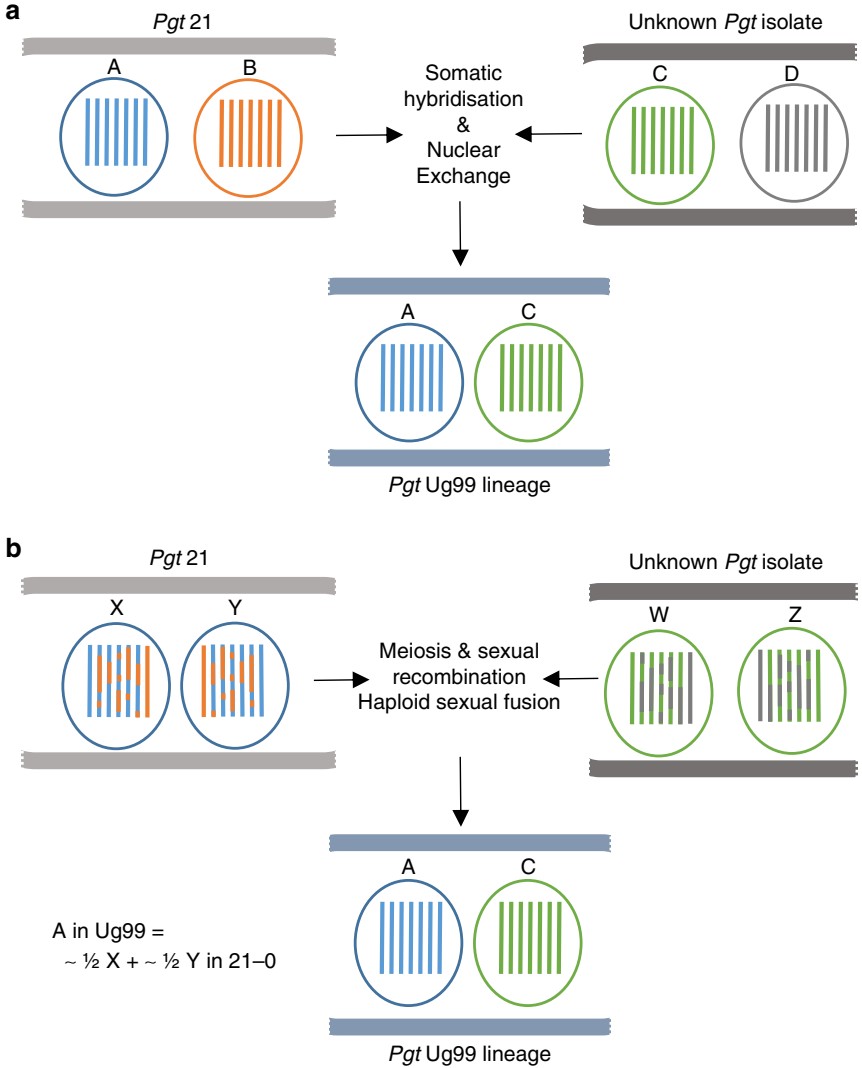

**Fig. 5** Models for the emergence of the founder isolate of the *Pgt* Ug99 lineage. **a** A somatic hybridisation event and nuclear exchange occurred between an isolate of the *Pgt* 21 lineage and an unknown *Pgt* isolate. The combination of nuclei A and C yielded the parental isolate of the Ug99 lineage in Africa. Under this scenario, nucleus A of Ug99 is entirely derived from nucleus A in *Pgt*21-0. **b** Alternatively, sexual reproduction and mating between these two parental isolates defined the origin of the Ug99 lineage. Under this scenario, meiotic recombination and chromosome reassortment would result in the *Pgt*21-0-derived A nucleus of Ug99 being composed of a mosaic of the two haploid nuclear genomes of *Pgt*21-0 (X and Y)

the A, B and C haplotypes were shared and represent a core *Pgt* gene set, while the remainder were present in only one or two haplotypes (Supplementary Table 8). Mapping of orthologous gene pairs supported the overall synteny of the *Pgt*21-0 A and B chromosome assemblies and confirmed the translocations observed between chromosomes 3 and 5, and between 8 and 16 (Fig. 8a). Genes encoding secreted and non-secreted proteins showed a similar distribution across the chromosomes, while repeat sequences displayed an inverse distribution to genes (Fig. 8b, Supplementary Fig. 5). The location of secreted protein genes in gene-rich rather than repeat-rich regions is consistent with the absence of two-speed genome architecture in the related rust fungal species *P. coronata* and *P. striiformis*[12,13].

Both Ug99 and *Pgt*21-0 are heterozygous at the predicted *a* and *b* mating type loci (Supplementary Fig. 6), despite Ug99 being derived by a non-sexual mechanism. This is consistent with an expectation that formation and maintenance of a stable dikaryon requires two distinct compatible mating types[11]. We observed multiple alleles at the *b* locus on chromosome 9, which encodes the divergently transcribed transcription factors bE and bW, with variants sharing 70–80% amino acid identity. Cuomo and colleagues[28] previously described two alleles, *b1* and *b2*, for this locus in *Pgt*. *Pgt*21-0 contained the *b1* allele and a novel *b3* allele, while Ug99 contained *b3* and another novel *b4* allele. Both isolates were heterozygous for the + and − alleles of the *a* locus on chromosome 4, which encodes a pheromone (mfa) and pheromone receptor (STE3) pair. However, one of the receptor alleles in Ug99 contained a single-nucleotide deletion that resulted in truncation of the last 48 amino acids of the protein. Thus, the mating type system for *Pgt* appears to consist of two independent loci, one di-allelic and one multi-allelic.

**Phylogenetic analysis of global *Pgt* isolates**. We used the haplotype-phased genome references for *Pgt*21-0 and Ug99 to determine genetic relationships within a set of global *Pgt* isolates

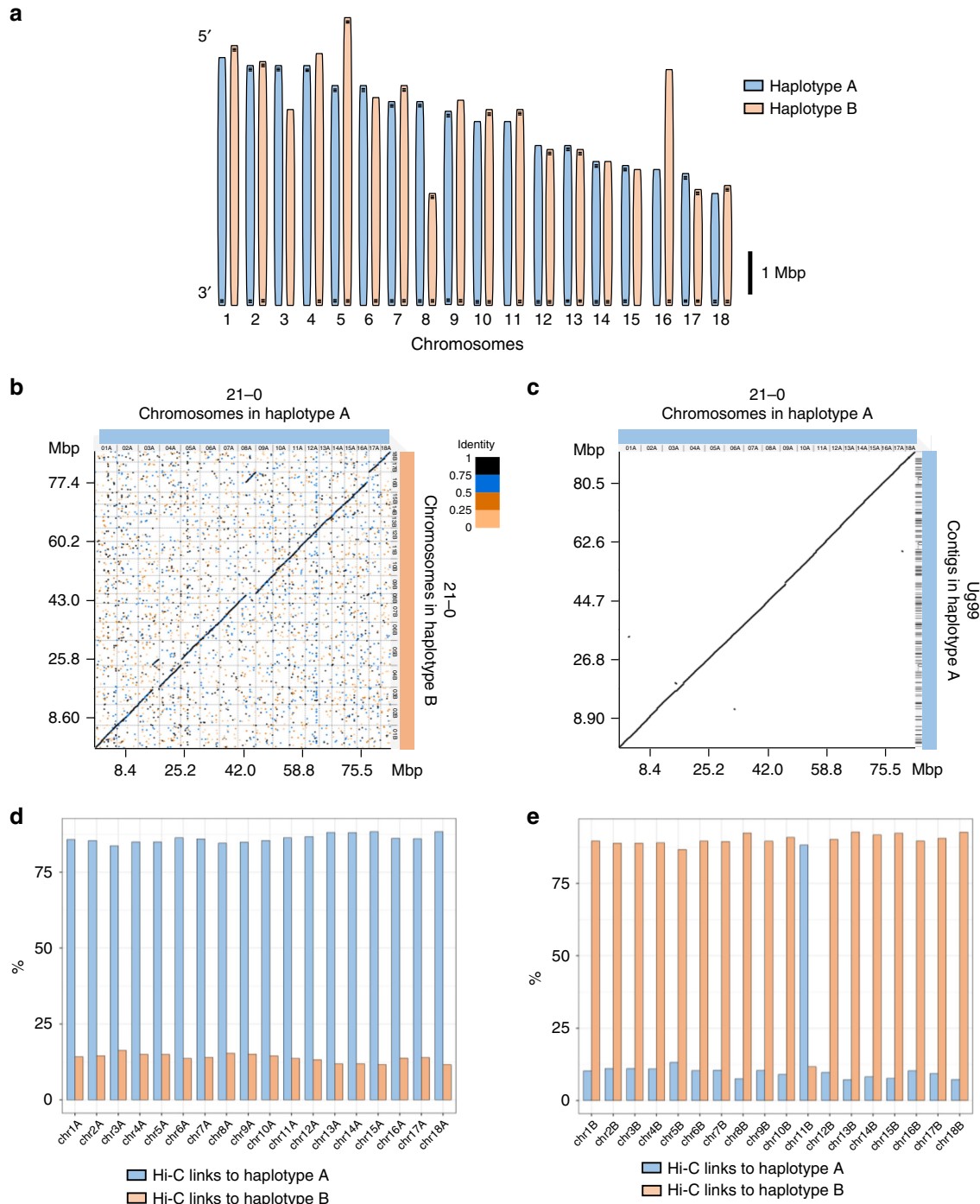

**Fig. 6** Chromosome sets of haplotype A and B in *Pgt*21-0. **a** Schematic representation of assembled chromosomes for *Pgt*21-0 of each haplotype (scale bar = 1 Mbp). Horizontal bars indicate telomeric repeat sequences. **b** Dot plot of sequence alignment of *Pgt*21-0 chromosome pseudomolecules of haplotypes A and B. Two translocation events, one between chromosomes 3 and 5 and one between chromosomes 8 and 16, are evident. **c** Dot plot of sequence alignment between chromosomes from haplotype A in *Pgt*21-0 and contigs from haplotype A in Ug99. **d** Percentage of Hi-C read pairs linking each A haplotype chromosome to other A chromosomes (blue) or to B haplotype chromosomes (orange). **e** Percentage of Hi-C read pairs linking each B haplotype chromosome to either A (blue) or B (orange) chromosomes

using publicly available sequence data[21,23,29]. Maximum likelihood trees based on whole-genome SNPs (Fig. 9a and Supplementary Fig. 7a) showed a very similar overall topology to that reported previously for most of these isolates[29]. The 5 isolates of the Ug99 lineage, and the 13 South African and Australian isolates each formed a separate tight clade, consistent with their proposed clonal nature[18–20]. However, tree building using filtered SNPs from just the A haplotype resulted in the formation of a single clade containing the Ug99, South African and Australian isolates, which indicates the clonal derivation of this nucleus among these isolates (Fig. 9b, Supplementary Fig. 7b, c). The Ug99 group forms a subclade within the race 21 group consistent with a derived origin. In contrast, these groups remained in two distant clades in phylogenies inferred using filtered SNPs from the B genome. Surprisingly, in this case two isolates from the Czech Republic and three isolates from Pakistan were now

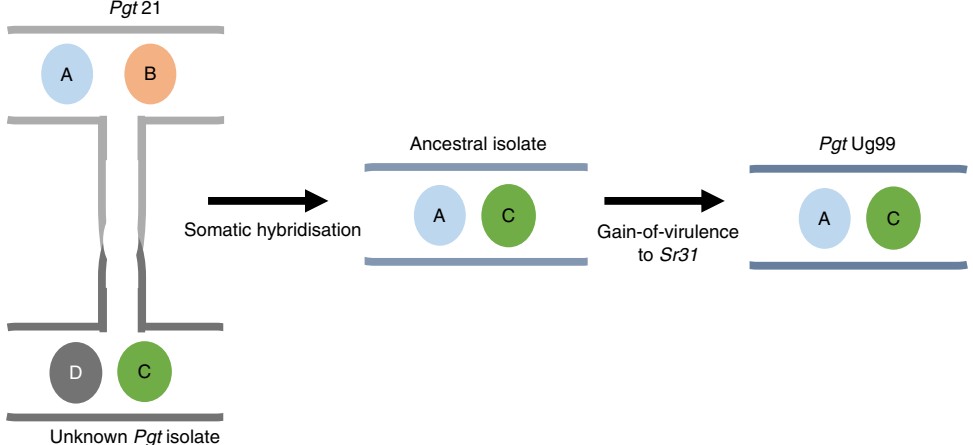

**Fig. 7** Model for Ug99 origin by somatic hybridisation and nuclear exchange. The ancestral isolate of the Ug99 lineage acquired the A and C genomes from an isolate of the 21 lineage and an unknown isolate and later gained virulence to wheat cultivars carrying the *Sr31* resistance gene

located in a single clade with the South African and Australian isolates (Fig. 9c). This suggests that these isolates contain a haplotype closely related to the B genome of the race 21 lineage and may also have arisen by somatic hybridisation and nuclear exchange. A phylogeny based on the C genome SNPs grouped isolate IR-01 from Iran with the Ug99 lineage (Fig. 9d), suggesting that these isolates share the C haplotype. IR-01 could represent a member of the parental lineage that donated the C nucleus to Ug99, or alternatively may have acquired the C nucleus from Ug99. Notably, this was the only isolate that shared the *AvrSr35* 57kbp insertion allele identified in Ug99 (Supplementary Fig. 8a). The relationships between these putative hybrid isolates were also supported by the patterns of homozygous and heterozygous SNPs detected in each haplotype (Supplementary Fig. 8b–e). The incongruities between phylogenies generated based on different haplotypes highlight the difficulty of inferring relationships between isolates based on whole-genome SNP data without haplotype resolution. Overall, these observations suggest that somatic hybridisation and nuclear exchange may be a common mechanism generating genetic diversity in global populations of *Pgt*.

## Discussion

Although sexual reproduction of *Pgt* can generate individuals with novel genetic combinations, the completion of the sexual cycle requires infection of an alternate host, *Berberis* spp. (barberry)[30]. In parts of the world where barberry is scarce or absent, either due to eradication programmes or its natural distribution, *Pgt* is restricted to asexual propagation with new diversity often arising by mutation or migration[19,20]. Somatic hybridisation provides an alternative explanation for the appearance of new races not derived by stepwise mutation. Hybrids with high adaptive value in agroecosystems may establish new lineages of epidemiological significance, as shown by the emergence of the Ug99 lineage with its substantial impact on East African wheat production and threat to global food security[14]. Results from experiments mostly conducted prior to the molecular biology era suggested the possibility of somatic genetic exchange between rust isolates co-infecting the same host, based on the detection of novel virulence phenotypes[5–8,31]. In some cases only two non-parental phenotypic classes were observed, consistent with simple nuclear exchange[7,8]. The isolation of additional recombinant classes in other experiments was interpreted as recombination between nuclei, although this was also ascribed to isolate contamination[7,32], which could not be ruled out without

molecular data. Our data show that Ug99 arose by a somatic nuclear exchange event with no recombination. Phylogenetic analyses show that at least two different global lineages (from Pakistan and the Czech Republic) share a haplotype similar to the *Pgt*21-0 B genome, while another (from Iran) shares a haplotype closely related to the Ug99 C genome. This suggests that multiple nuclear exchange events between strains have occurred in the global *Pgt* population and had significant impacts on genetic diversity. Although we did not observe any recombination between nuclei associated with the Ug99 hybridisation event, we did see evidence for translocation of one complete chromosome between nuclei in *Pgt*21-0. We also previously found that a *Pgt* mutant virulent on *Sr50* arose by exchange of an ~2.5 Mbp region between two haplotypes[23]. Thus, genetic exchange between haploid nuclei may occur as rare events during asexual propagation of a single lineage in *Pgt*. Whether extensive genetic exchange similar to ascomycete parasexuality[2] can also occur between rust nuclei during hybridisation remains to be determined. This may require controlled infection experiments, as such recombinant hybrids would be difficult to distinguish from the products of sexual recombination in field derived-strains, especially given the potential for long-range spore dispersal.

Although there is now clear evidence of nuclear exchange between dikaryons in *Pgt*, nothing is known of how this process occurs or is regulated. It differs from parasexuality in ascomycetes[2], as the dikaryotic state is maintained with no nuclear fusion or haploidisation resulting in chromosome reassortment. Wang and McCallum[33] observed the formation of fusion bodies where germ tubes of different *P. triticina* isolates came into contact, with the potential for nuclear exchange at these junctions. It has been proposed that mating type loci contribute to determining the compatibility between isolates for the formation of hybrids[6,7], but this also remains to be confirmed experimentally. Our findings provide a new framework to take advantage of haplotype genome resolution to understand the role of somatic exchange in population diversity of rust fungi.

Extended dikaryotic developmental stages are common in many other fungi, especially basidiomycetes. Indeed, separation of karyogamy (fusion of haploid nuclei to form a diploid nucleus) from gamete fusion is a feature unique to the fungal kingdom[1]. However, it is unclear why fungi maintain an extended dikaryotic stage prior to formation of a diploid nucleus as a precursor to sexual reproduction[34]. One possibility is that the ability to exchange haploid nuclei offers an advantage over the diploid state due to the enhanced genetic variation in long-lived asexual

**a**

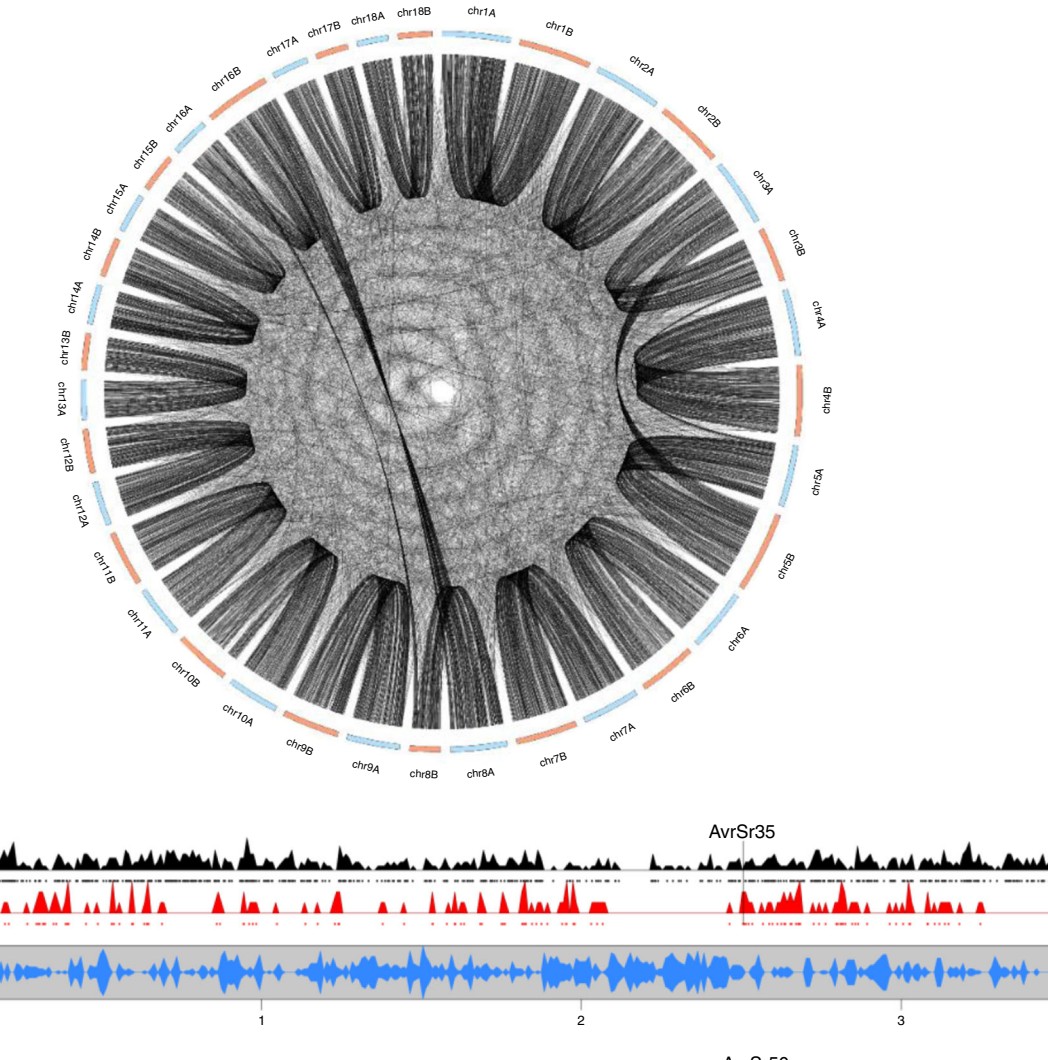

**b**

**Fig. 8** Gene content of *Pgt*21-0 chromosome pseudomolecules. **a** Circos plot showing location of orthologous gene pairs in the A and B chromosomes of *Pgt*21-0. **b** Gene and repeat density plots for homologous chromosomes 14 A and 14B. Density of genes encoding non-secreted (black) or secreted proteins (red) along the chromosomes are shown, with individual genes indicated by black or red dots. Bottom graph shows density of repeat elements (blue). Positions of *AvrSr50* and *AvrSr35* genes are indicated

dikaryotes. There is also evidence for somatic exchange of genetic markers in dikaryotes of the mushroom *Schizophyllum commune*, which belongs to another Basidiomycete subphylum, Agaricomycotina[35]. Arbuscular mycorrhizae (AM) fungi are another ancient fungal lineage whose spores contain hundreds of nuclei and for which no sexual stages have been described, raising questions of how these lineages have survived[36]. Recently some dikaryotic-like AM isolates possessing two divergent classes of nuclei have been observed. Nuclear exchange between dikaryotes could be another driver of genetic variation in these fungi. Evidently, the members of the fungal kingdom display remarkable genetic plasticity and further investigation is required to reveal the mechanism, prevalence and evolutionary importance of nuclear exchange in dikaryotic and multinucleate fungi.

## Methods

**Fungal stocks and plant inoculation procedures**. *Pgt* isolates Ug99[15], UVPgt55, UVPgt59, UVPgt60 and UVPgt61 collected in South Africa[16,37] were transferred to the Biosafety Level 3 (BSL-3) containment facility at the University of Minnesota for growth and manipulation. Samples were purified by single pustule isolation and then amplified by 2–3 rounds of inoculation on the susceptible wheat cultivar McNair. Virulence pathotypes and purity of each isolate were confirmed by inoculation onto the standard wheat differential set (Supplementary Data 1). Other isolates used in this study were *Pgt*21-0, which was first isolated in Australia in 1954[20,21], the North American isolate CRL 75-36-700-3 (pathotype SCCL)[22] and Kenyan isolate 04KEN156/04 (pathotype TTKSK)[17]. For rust inoculations, urediniospores retrieved from −80 °C were heat treated (45 °C for 15 min) and suspended in mineral oil (Soltrol 170, Philips Petroleum, Borger, TX, USA) at 14 mg/ml. Seven day-old seedlings were spray-inoculated at 50 μl/plant and kept in a dark mist chamber at 22–25 °C, 100% humidity for 16 h. Subsequently, plants were exposed to light (150–250 μmol photons s$^{-1}$ m$^{-2}$) for 5 h in the mist chamber and then transferred to a controlled growth chamber (18 h/6 h of light/dark, 24 °C/

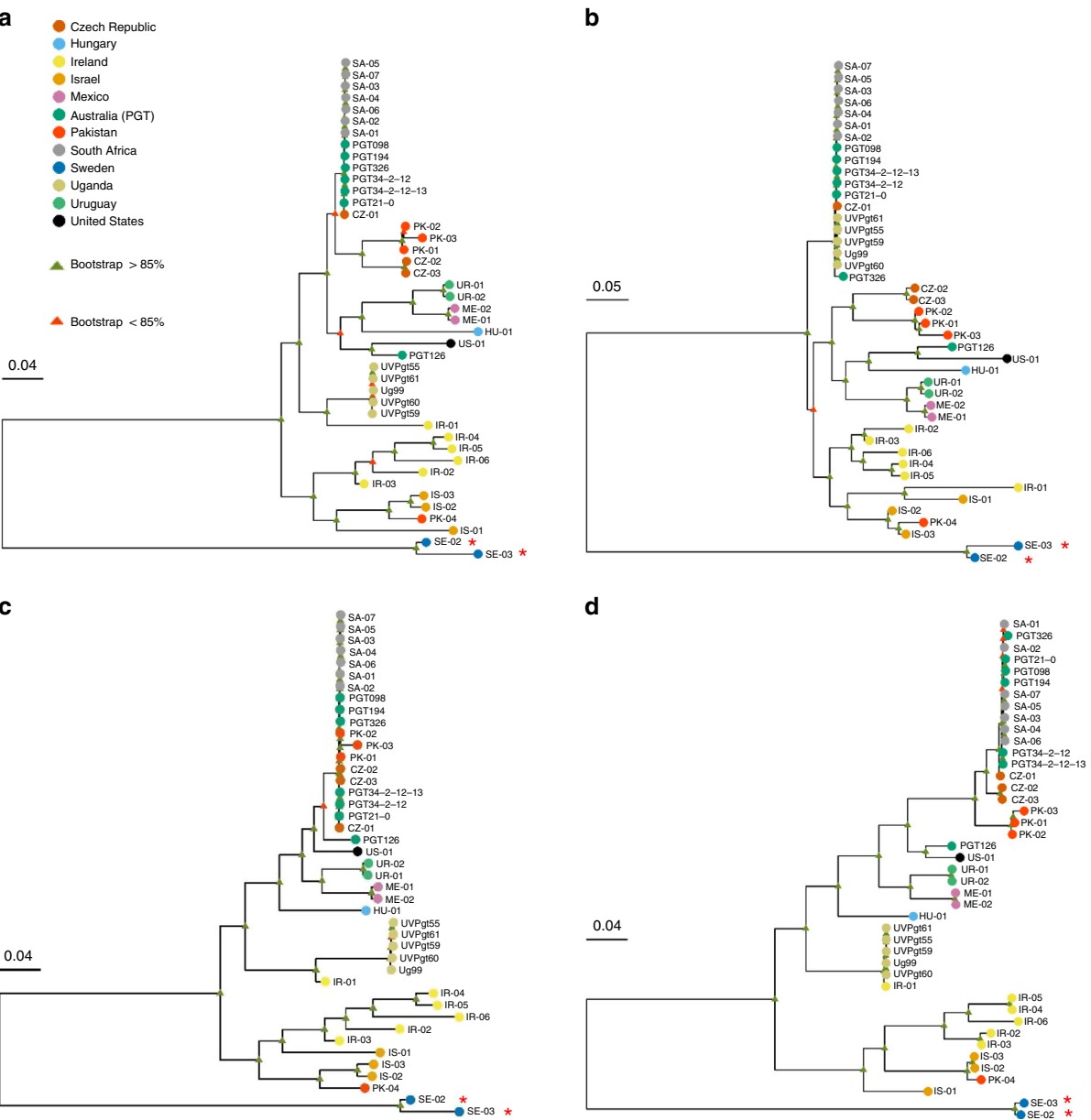

**Fig. 9** Somatic hybridisation in *Pgt* evolution. **a** Phylogenetic analysis of *Pgt* isolates from diverse countries of origin (colour key) using a RAxML model and biallelic SNPs called against the full dikaryotic genome of *Pgt*21-0. Scale bar indicates number of nucleotide substitutions per site. Red asterisks indicate *P. graminis* f. sp. *avenae* isolates used as an outgroup. **b** Dendrogram inferred using biallelic SNPs detected against haplotype A of *Pgt*21-0. **c** Dendrogram inferred using SNPs detected against haplotype B of *Pgt*21-0. **d** Dendrogram inferred from SNPs detected in haplotype C of Ug99

18 °C for day/night, 50% relative humidity). Spores were collected 9 and 14 days post inoculation (dpi) and maintained at −80 °C.

**DNA extraction and sequencing of rust isolates**. High molecular weight DNA of Ug99 and *Pgt*21-0 was extracted from 300 to 350 mg urediniospores as described[38], with the following modifications: (1) for Phenol:Chloroform:Isoamyl alcohol extractions, samples were centrifuged at 4 °C and 5000 × *g* for 20 mins; (2) a wide-bore 1-mL pipette tip was used to transfer the DNA pellet; (3) samples were incubated for 1 h at 28 °C with 200–250 rpm shaking to dissolve the final DNA pellet. Double-stranded DNA concentration was quantified using a broad-range assay in a Qubit Fluorometer (Invitrogen, Carlsbad, CA, USA) and a NanoDrop (Thermo Fisher Scientific, Waltham, MA, USA). Approximately 10 μg DNA from Ug99 and *Pgt*21-0 was sequenced using PacBio SMRT sequencing (Pacific Bioscience, Menlo Park, CA, USA) at either the Frederick National Laboratory for Cancer Research, Leidos Biomedical Research, Inc. (Frederick, MD, USA) or the Ramaciotti Centre (Sydney, Australia), respectively. DNA was concentrated and cleaned using AMPure PB beads for Ug99 or AMPure XP beads for *Pgt*21-0 (Pacific Biosciences). DNA quantification and size assessment were conducted using a NanoDrop (Thermo Fisher Scientific) and 2200

TapeStation instruments (Agilent Technologies, Santa Clara, CA, USA). DNA was sheared to a targeted average size of 20 kb using G-tubes (Covaris, Woburn, MA, USA). Libraries were constructed following the 20 kb Template Preparation BluePippin Size-Selection System protocol (Pacific Biosciences) using a Blue-Pippin instrument (Sage Science, Beverly, MA, USA) with a 0.75% agarose cassette and a lower cutoff of 15 kbp. For Ug99, 5 SMRT cells were sequenced on a PacBio Sequel platform using P6-C4 chemistry, the Sequel Binding Kit 2.0 (Pacific Biosciences), diffusion loading, 10-h movie lengths and Magbead loading at 2 pM (3 cells) or 4 pM (2 cells). In addition, 4 SMRT cells were run on PacBio RSII sequencer using P6-C4 chemistry, with 0.15 nM MagBead loading and 360-min movie lengths. For *Pgt*21-0, 17 SMRT cells were run on the RSII plat-form using P6-C4 chemistry, Magbead loading (0.12–0.18 nM) and 240-min movie lengths.

Genomic DNA for Illumina sequencing was extracted from 10 to 20 mg urediniospores of Ug99, UVPgt55, 59, 60 and 61 using the OmniPrep™ kit (G-Biosciences, St. Louis, MO, USA) following the manufacturer's instructions. TruSeq Nano DNA libraries were prepared from 300 ng of DNA and 150 bp paired-end sequence reads were generated at the University of Minnesota Genomics Center on the Illumina NextSeq 550 platform using Illumina Real-Time Analysis software version 1.18.64 for quality-scored base calling.

**De novo long read assembly**. Genome assemblies of Ug99 and *Pgt*21-0 were built from PacBio reads using Canu version 1.6[39] with default parameters and an estimated genome size of 170 Mbp. Assemblies were polished with the Arrow algorithm using the raw PacBio reads in the sa3_ds_resequencing pipeline in pbsmrtpipe workflow within SMRTLINK/5.1.0 (Pacific BioSciences). Assemblies were further polished by two rounds of Pilon[40] with the option fix --all using Illumina reads from Ug99 (this work) or *Pgt*21-0 (NCBI SRA run Accession# SRR6242031). A BLASTN search (version 2.7.1) against the NCBI nr/nt database (downloaded on 4/11/2018) with E-value set as $1e^{-10}$ identified two contigs in the Ug99 assembly with significant hits to plant rRNA and chloroplast sequences and these were removed.

PacBio and Illumina reads were mapped to the assembly using BWA-MEM (version 0.7.17)[41] and BAM files were indexed and sorted using SAMtools (version 1.9)[42]. Read coverage analysis using genomeCoverageBed in BEDtools (version 2.27.1)[43] identified 144 small contigs (<50 kbp) in the Ug99 assembly with low coverage (<2×) for both short and long reads and these contigs were also excluded from the final assembly. Genome assembly metrics were assayed using QUAST (version 4.3)[44]. Genome completeness was assessed via benchmarking universal single-copy orthologs (BUSCOs) of the basidiomycota as fungal lineage and *Ustilago maydis* as the reference species for AUGUSTUS gene prediction[45] in the software BUSCO v2.0 (genome mode)[46]. Telomeric sequences were identified using either a high stringency BLAST with 32 repeats of TTAGGG as query or a custom python script to detect at least five CCCTAA or TTAGGG repeats in the assemblies. Repeats of at least 60 bp length and occurring within 100 bp of the contig end were defined as telomeric sequences.

**Detection of alternate contigs and bin assignment**. To identify contigs representing corresponding haplotypes (Fig. 1) 22,484 predicted *Pgt* gene coding sequences[21] were screened against the genome assemblies using BLITZ (Blat-like local alignment) in the Biokanga Tool set, (https://github.com/csiro-crop-informatics/biokanga/releases/tag/v4.3.9). For each gene the two best hits (likely alleles) in the assembly were recorded. Contigs sharing best hits for at least five genes were selected as potential haplotype pairs and their sequence collinearity was examined by alignment and similarity plotting using D-genies[47]. Contigs representing contiguous or syntenous haplotypes were grouped together as bins.

**Validation of a 57-kbp-insert in *AvrSr35***. Contigs containing the *AvrSr50* and *AvrSr35* gene sequences were identified by BLASTN search against customised databases for the Ug99 and *Pgt*21-0 genome assemblies. Illumina and PacBio reads of Ug99 mapped to the genome assembly were visualised in the Integrative Genomics Viewer (IGV). To validate the presence of the 57-kbp-insert in *AvrSr35*, flanking and internal sequences were amplified from genomic DNA extracted using the OmniPrep™ kit (G-Biosciences) from urediniospores of Ug99, 04KEN156/04, and CRL 75-36-700-3. PCR was performed using Phusion high-fidelity DNA polymerase according to the manufacturer's recommendations (New England BioLabs Inc., Ipswich, MA, USA) and primers listed in Supplementary Table 9. The amplified PCR products were separated by electrophoresis on a 1% agarose gel and stained using SYBR Safe DNA gel stain (Invitrogen). Specific bands were cleaned using NucleoSpin gel clean-up kit (Takara Bio, Mountain View, CA, USA) for subsequent Sanger sequencing and alignment to *AvrSr35* alleles. Gene models in the *AvrSr35* and *AvrSr50* locus were depicted using GenomicFeatures[48] and ggbio[49] in a custom R script.

**Haplotype assignment by read cross-mapping and subtraction**. Illumina reads from *Pgt*21-0 (NCBI SRR6242031) were trimmed ("Trim sequences" quality limit = 0.01) and mapped to the Ug99 reference assembly using the "map reads to reference" tool in CLC Genomics Workbench version 10.0.1 or later with high stringency parameters (similarity fraction 0.99, length fraction 0.98, global alignment). Unmapped reads (Ug99-subtracted reads) were retained and then mapped back to the *Pgt*21-0 assembly contigs using the same parameters. The original *Pgt*21-0 reads were also mapped to the *Pgt*21-0 assembly and the read coverage for each contig compared to the Ug99-subtracted reads. Contigs with very low coverage (<2X total and <10% of the original read coverage) with the Ug99-subtracted reads were designated as karyon A (Fig. 3, Supplementary Data 3). Contigs with substantial coverage of Ug99-subtracted reads (>10% of the original read coverage) were designated as karyon B. Contigs with ambiguous read mapping data, including those with low coverage in the original unsubtracted reads or covered by largely non-uniquely mapping reads were left as unassigned. Read mapping to all contigs was confirmed by visual inspection of coverage graphs and read alignments in the CLC Genomics Workbench browser. Potential chimeric contigs were identified as containing distinct regions with either high or no coverage with the Ug99-subtracted reads (Supplementary Fig. 4). For subsequent comparison and analyses, these contigs were manually split into their component fragments which were designated as haplotype A or B accordingly (Supplementary Data 4). The same process was followed in reverse for the assignment of the A and C haplotype contigs in Ug99.

**Sequence comparisons of genome assemblies**. Haplotype sequences of the *AvrSr50/AvrSr35* chromosome as well as the full haploid genomes were aligned

using MUMmer4.x[50] with nucmer -maxmatch and other parameters set as default and the alignment metrics summarised in MUMmer dnadiff. Structural variation between haplotypes was determined using Assemblytics[50] from the MUMmer delta file with a minimum variant size of 50 bp, a maximum variant size of 100 kbp, and a unique sequence length for anchor filtering of 10 kbp. Haplotype dot plot alignments were generated using D-genies[47].

**Read coverage analysis and SNP calling on haplotypes**. Illumina reads from Ug99 and *Pgt*21-0 were each mapped against the Ug99 and *Pgt*21-0 assemblies in CLC Genomics Workbench (similarity fraction 0.98, length fraction 0.95). For each assembly the mean coverage per base was calculated per 1000 bp interval ("window") using samtools bedcov. Read coverage frequency normalised to the mean coverage of each haplotype was graphed as a violin plot using seaborn 0.9.0 package (https://seaborn.pydata.org/) using a custom python script. To detect SNPs between two haplotypes, Illumina read pairs of *Pgt*21-0 that mapped uniquely to either the *Pgt*21-0 A or B haplotype contigs were extracted. Similarly, Ug99-derived read pairs that uniquely mapped to either the A or C haplotype contigs of Ug99 were extracted. These read sets were then separately mapped to the two assemblies in CLC Genomics Workbench (similarity fraction 0.99, length fraction 0.98). Variant calling was processed with FreeBayes v.1.1.0[51] with default parameters in parallel operation and SNPs were filtered using vcffilter of VCFlib (v1.0.0-rc1, https://github.com/vcflib/vcflib) with the parameter -f "QUAL > 20 & QUAL / AO > 10 & SAF > 0 & SAR > 0 & RPR > 1 & RPL > 1". Homozygous and heterozygous SNPs were extracted by vcffilter -f "AC > 0 & AC = 2" and -f "AC > 0 & AC = 1", respectively. SNP statistics were calculated using vcfstats of VCFlib.

**Hi-C data analysis and scaffolding**. A Hi-C library was constructed with the ProxiMeta Hi-C kit from Phase Genomics v1.0 containing the enzyme Sau3A from ~150 mg of dried urediniospores of *Pgt*21-0 following the standard protocol with minor modifications. Spores were washed in 1 mL 1X TBS buffer twice before cross-linking. After quenching of the crosslinking reaction, all liquid was removed and spores were frozen in liquid nitrogen. Spores were then lysed using cryogenic bead beating with two 5 mm steel beads shaking twice for 45 sec at 25 Hz using TissueLyser II (Qiagen). Lysis buffer was added and spores vortexed until full suspension. Reverse cross-linking was performed at 65 °C with 700 rpm horizontal shaking for 18 h. The Hi-C library was sequenced (150 bp paired-end reads) on the NextSeq 550 System using the Mid-Output Kit at the Ramaciotti Centre. The raw Hi-C reads were processed with the HiCUP pipeline version 0.7.1[52] (maximum di-tag length 700, minimum di-tag length 100, --re1 ^GATC,Sau3A), using bowtie2 as the aligner[53] and the *Pgt*21-0 genome assembly as the reference. SAM files of the filtered di-tags were parsed to extract cis-far pairs (pairs located on the same contig and > 10 kbp apart) and trans pairs (located on different contigs). The numbers of trans pairs connecting each pair of contigs was extracted from this data. We compared the number of Hi-C read pair connections between contigs of the same or different haplotypes, either within bins, within chromosomes or at the whole-genome level (all haplotype-assigned contigs). We used a $X^2$ test to assess the deviation of each read pair distribution from a 1:1 ratio.

For scaffolding, the raw Hi-C reads were first mapped to the *Pgt*21-0 assembly using BWA-MEM[41] and processed using the Arima Genomics pipeline (https://github.com/ArimaGenomics/mapping_pipeline/blob/master/01_mapping_arima.sh). Scaffolding was performed using SALSA 2.2[54] on the full set of contigs, as well as independently on the haplotype A or B sets of contigs (including unassigned contigs). Invalid scaffold linkages between adjacent telomeres, which occur as an artefact of telomere co-location within the nucleus, were discarded. The three sets of scaffolds were compared with the bin and haplotype assignment information to find overlaps and the resulting chromosome pseudomolecules were constructed by concatenating ordered contigs with 100 Ns inserted between contigs. To confirm translocations detected in the A and B chromosome sets, contigs that spanned the translocation site were separated into two fragments at the junction point and the SALSA scaffolding process was repeated on the full genome contig assembly. To detect nucleus-specific cross-links between chromosomes, HiCUP analysis was performed using the chromosome pseudomolecules as the reference assembly and the proportion of trans linkages between chromosomes of the same or different haplotype computed.

**Gene prediction and functional annotation**. The genome assemblies of Ug99 and *Pgt*21-0 (as chromosome pseudomolecules for *Pgt*21-0) were annotated using the Funannotate pipeline (https://github.com/nextgenusfs/funannotate). Contigs were sorted by length (longest to shortest) and repetitive elements were soft-masked using RepeatModeler (v1.0.11) and RepeatMasker (v4.0.5; http://www.repeatmasker.org/) with RepBase library (v. 23.09)[55]. RNAseq libraries from *Pgt*21-0 (Supplementary Data 7)[21,23] were used for training gene models. In the training step, RNAseq data were aligned to the genome assembly with HISAT2[56]. Transcripts were reconstructed with Stringtie (v1.3.4d)[57]. Genome-guided Trinity assembly (v2.4.0)[58] and PASA assembly (v2.3.3)[59] were performed. To assist in predicting effector-like genes, stringtie-aligned transcripts were used in CodingQuarry Pathogen Mode (v2.0)[60]. The prediction step of funannotate pipeline (funannotate predict) was run with --ploidy 2, --busco_db basidiomycota and default parameters. Transcript evidence included Trinity transcripts, Pucciniamycotina EST clusters downloaded

from the JGI MycoCosm website (http://genome.jgi.doe.gov/pucciniomycotina/pucciniomycotina.info.html, April 24, 2017), and predicted transcript sequences of haustorial secreted proteins[21]. Transcript evidence was aligned to the genome using minimap2 v2.1.0[61] and the protein evidence was aligned to genome via Diamond (v0.9.13)/Exonerate (v2.4.0)[62] using the default UniProtKb/SwissProt curated protein database from funannotate. Ab initio gene predictor AUGUSTUS v3.2.3[45] was trained using PASA data and GeneMark-ES v4.32[63] was self-trained using the genome assembly. Evidence Modeler was used to combine all the above evidence using default weight settings except that the weight of PASA and CodingQuarry_PM were both set to 20. tRNA genes were predicted using tRNAscan-SE v1.3.1[64]. Gene models including UTRs and alternative spliced transcripts were updated using RNAseq data based on Annotation Comparisons and Annotation Updates in PASA. Funannotate fix was run to validate gene models and NCBI submission requirements. Genome annotation was assessed using BUSCO v2.0 (transcript and protein modes)[46]. For functional annotation protein coding gene models were firstly parsed using InterProScan5 (v5.23-62.0) to identify InterPro terms, GO ontology and fungal transcription factors[65]. Pfam domains were identified using PFAM v. 32.0, and carbohydrate hydrolysing enzymatic domains (CAZYmes) were annotated using dbCAN v7.0[66]. Diamond blastP[67] was used to search UniProt DB v. 2018_11[68] and MEROPS v. 12.0[69] databases to aid in functional annotation. BUSCO groups were annotated with Basidiomycota models and eggNOG terms were identified using eggNOG-mapper v1.0.3[70]. Putative mating-type loci in Pgt21-0 and Ug99 were identified by BLAST search with the pheromone peptide encoding genes (mfa2 or mfa3) and pheromone mating factor receptors (STE3.2 and STE3.3) from the a locus, and bW/bE transcription factors from the b locus previously identified in Pgt[28]. Protein sequences were aligned in Clustal Omega[71].

Gene and repeat density plots for chromosomes were generated using karyoploteR[72]. Secreted proteins were predicted using the neural network predictor of SignalP 3.0[73] and retained if they lacked a transmembrane domain outside the first 60 amino acids using TMHMM[74]. RepeatMasker 4.0.6 with the species fungi was used to softmask repeats. Repeats longer than 200 bp were used in the chromosome plotting.

**Orthology analysis**. Gene annotations with multiple isoforms were reduced to a representative isoform by selecting the longest CDS using a custom perl script. Orthologous proteins were identified with Orthofinder[75] using default parameters. Multiple pairwise orthology analyses were run based on within-isolate and cross-isolate comparisons of similar haplotypes (i.e. A versus A or B versus C). Additional comparisons were made between Pgt21-0 A, Pgt21-0 B, and Ug99 C haplotypes, as well as between Ug99 A, Ug99 C and Pgt21-0 B haplotypes.

**Phylogenetic analysis of rust isolates**. For whole-genome SNP calling and phylogenetic analysis we used Illumina DNA sequence data from the five Ug99 lineage isolates described here, seven Australian isolates[21,23] as well as 31 global isolates[29] downloaded from the European Nucleotide Archive (ENA; PRJEB22223) (Supplementary Data 8). Read quality was checked using FASTQC (http://www.bioinformatics.babraham.ac.uk/projects/fastqc/) and reads were trimmed with Trimmomatic v.0.33[76] using default settings and reads <80 bp were discarded. Trimmed reads were aligned to the Ug99 or Pgt21-0 genome assemblies using BWA-MEM v.0.7.17[41] and technical replicates were merged using SAMtools 1.6[42] and the PICARD toolkit (http://broadinstitute.github.io/picard/). Read lengths and coverage were verified by the functions bamtobed and coverage in BEDtools[43] and flagstat in SAMtools. FreeBayes v. 1.1.0[51] was used to call biallelic SNP variants across the 43 samples simultaneously. VCF files were filtered using vcffilter in vcflib (v1.0.0-rc1) with the parameters: QUAL > 20 & QUAL / AO > 10 & SAF > 0 & SAR > 0 & RPR > 1 & RPL > 1 & AC > 0. To verify that each sample consisted of a single genotype free of contamination, read allele frequencies at heterozygous positions were examined using the vcfR package[77]. VCF files were converted to multiple sequence alignment in PHYLIP format using the vcf2phylip script (https://zenodo.org/record/1257058#.XNnE845Kh3g) and R-package ips/phyloch wrappings (http://www.christophheibl.de/Rpackages.html). Phylogenetic trees were constructed using the maximum likelihood criterion in RAxML v. 8.2.1.pthread[78], assuming unlinked loci and using 500 bootstrap replicates with a general time reversible model. Convergence and posterior bootstopping (bootstrapping and convergence criterion) were confirmed with the -I parameter in RAxML and also with R-packages ape[79], ips/phyloch, and phangorn[80]. Dendrograms were drawn using ggplot2[81] and ggbio[49] R-packages.

SNPs called against the A, B or C haplotypes were separated from the total SNP sets using the function intersect –header in BEDtools. Homozygous and heterozygous SNPs were extracted by vcffilter -f "TYPE = snp" and -f "AC > 0 & AC = 2" and -f "AC > 0 & AC = 1", respectively and their frequency was counted using vcfkeepsamples and vcffixup. SNP statistics were calculated using vcfstats of VCFlib (v1.0.0-rc1).

**Reporting summary**. Further information on research design is available in the Nature Research Reporting Summary linked to this article.

## Data availability

All sequence data, assemblies and gene annotation files generated in this study are available in NCBI under BioProject PRJNA516922. Assemblies and annotations are also available at the DOE-JGI Mycocosm Portal (https://mycocosm.jgi.doe.gov/mycocosm/home). Metadata for RNAseq libraries of Pgt21-0 and Illumina DNAseq libraries from all isolates are available in Supplementary Data 7 and 8, respectively. Data underlying Fig. 3c and Supplementary Fig. 3a, b is provided in Supplementary Data 3. Data underlying Fig. 6 is available in Supplementary Data 6. All other relevant data is available upon request from the corresponding authors.

## Code availability

Unless specified otherwise, all scripts and files are available at https://github.com/figueroalab/Pgt_genomes.

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

## Acknowledgements

We thank J. Ellis, P. van Esse, G. Bakkeren, C. Aime and Y. Jin for valuable discussions, S. Dahl and N. Prenevost for technical support, J. Palmer for gene annotation trouble-shooting, and the Minnesota Supercomputing Institute for computational resources. This research was funded by two independent grants from the 2 Blades foundation to P.N.D. and M.F., respectively, by a USDA-Agriculture and Food Research Initiative (AFRI) Competitive Grant (Proposal No. 2017-08221) to M.F, and University of Minnesota Lieberman-Okinow Endowment to B.J.S.; M.F. and M.E.M were supported by the University of Minnesota Experimental Station USDA-NIFA Hatch/Figueroa project MIN-22-G19 and an USDA-NIFA Postdoctoral Fellowship award (2017-67012-26117), respectively. B.S. is supported by an ARC Future Fellowship (FT180100024). J.S. is supported by an ARC DECRA Fellowship (DE190100066).

## Author contributions

M.F. and P.N.D. conceptualised the project, acquired funding and supervised the work. B.V. and Z.A.P. provided study materials. F.L., N.M.U., C.R., O.M., B.S., R.M. and B.J.S. acquired experimental data. F.L., N.M.U., J.S., B.S., B.J.S., H.N.P., P.N.D, K.A.T.S., E.H., M.E.M. and C.D.H. conducted data analysis. M.F., P.N.D. and F.L. drafted the manuscript. All authors contributed to review and editing.

## Competing interests

The authors declare no competing interests.
