## [Peer Review File · Nature Communications]

Reviewers' comments:

Reviewer #1 (Remarks to the Author):

Review of Li et al NComms

This report by Li et al describes the use of haplotype phased genome assemblies for two isolates of the wheat stem rust fungus *Puccinia graminis* to infer a historical hybridization event. The authors use PacBio reads to achieve highly contiguous, phased contigs, and combine this with Hi-C linking reads to infer chromosome organization. Comparing the two assemblies revealed that both share one haplotype at high identity, and confirm that this consists of chromosomes from a single nucleus with one exception. The second haplotype shows much lower sequence similarity to the first and between the two isolates.

Overall, the authors present an exciting story of hybridization in Pgt supported by state of the art genomic data and approaches. My main comment would be a concern that too much has been distilled down to examples and diagrams, with the bulk of the data analysis not depicted in the main text figures and tables. While this journal has a general audience, I found this to be a difficult read in places in needing to jump back and forth to understand what was done. However, these are minor issues in presenting the work behind an important piece of genomic detective work for this field. A larger question for future work is how this hybridization event may have led to the emergence of Ug99 in particular as a highly virulent lineage.

Minor comments:

Line 71: The call out here for Supplementary Table 1 is unclear as there is no mention of virulence pathotypes. Please add the name/number of this supplementary table to the header of the file (also Table S5), as when downloading in bulk it is not clear which table this is.

Line 72: In describing the assembly, add the contig counts and N50 to give better context to the synteny mapping that follows.

Line 75: Are the "conserved fungal genes" the BUSCO set shown in Figure 3d?

Line 77: Explain the equation $n=123$

Line 77-80: How do the bins compare to chromosome number?

Figure 2- I found this figure difficult to follow, to understand where the insertions are relative to the gene models. It would help to only have a single frame of reference for depicting the gene structures- the arrows being a different size than the track with coordinates complicated this image. Does AvrSr50 include several exons, or is that representing an insertion in that region?

Line 98: Here the authors note very low identity between the haplotypes; why are these values so much lower than the identity reported at line 135? From Supplementary Table 6 it appears this is due to considering non-aligning regions in calculating identity- was that also the case for the calculations at line 135?

Figure 3a and b- Can the authors show plots of all the subtracted read alignment data to illustrate how the coverage thresholds were used to select the haplotypes? These example diagrams while clear provide idealized illustrations of the approach.

Line 111- the text on conserved fungal genes refers to Figure 3c; this data is showing in Figure

3d.

Line 112- the text states that "the haplotypes were highly contiguous" referring to 3d, which is the BUSCO data.

Line 112-5- For the text on alignment identity here, the methods describe calculating identity from mummer whole genome alignments; this would not include a correction for regions that did not align as used for the Avr regions?

Figure 5- In the context of this model, is there supporting data on the level of recombination in Pgt during the sexual stage of its lifecycle? Has LD been estimated previously; this could also be done from the population genomic data.

Figure 6- In the figure legend, the descriptions of panels d and e are swapped.

L144: What significance test was used to evaluate Hi-C read mapping in Table S10?

L168: The statement about "no recombination events" may need some clarification. Were no SVs detected between the two A genomes? There are some off diagonal alignment segments show in Figure 4a- was there a size threshold (ie the 5 gene block used for haplotype analysis) to detecting rearrangements, such that small events may not have been counted?

L177: It appears that the ortholog mapping also supports the translocations described at line 154.

The section on phylogenetic analysis starting on line 183 refers to Figure 7, however this data is in figure 8. There also appears to be a discrepancy in the figure legend. The text notes a very similar clade of isolates using the A genome referring to Fig 7b (line 197). In Figure 8, the legend stated that panel 8b has the A genome, however this does not contain a very similar clade- that data appears to be in panel 8c.

The sequence data, both the raw data and the annotated assemblies, needs to be made available in NCBI. PRJNA516922 currently returns no results.

Reviewer #2 (Remarks to the Author):

In this well-written manuscript, the authors provide clear and convincing evidence for a role of parasexuality and somatic hybridisation in dikaryotic Basidiomycete fungi. Although well-known from haploid, monokaryotic, Ascomycete fungi, this process has not been shown to operate in Basidiomycetes. With their findings, the authors convincingly demonstrate how the Ug99 lineage of the wheat stem rust pathogen, *Puccinia graminis* f. sp. *tritici* evolved. The approach that was used by the authors is sound, state-of-the-art and elegant, and combines genome sequencing with HiC scaffolding, resulting in haplotype phased genome assemblies. Their conclusions are well-supported by the data.

I appreciate the work that the authors have performed and enthusiastically support publication of this manuscript. My only, minor, concern would be to suggest the authors to perhaps re-consider their final statement in the abstract where they state: "Generation of genetic variation by nuclear exchange may favour the evolution of dikaryotism by providing an advantage over diploidy". Although carefully phrased, the authors seem to suggest that one or the other, dikaryotism or diploidy, may be better or "preferred", while I do not think that evolution necessarily works towards the domination of one system. Clearly, sexual reproduction has advantages, but also comes at a cost. Throughout evolution, several mechanisms evolved to compensate for the lack of advantages of sexual reproduction in case asexual reproduction became the norm. As such,

generation of genetic variation by nuclear exchange may just be a means to compensate for loss of sex, although it needs to be noted that plenty Ptg lineages are sexually active, in case the barberry host is available. Finally, perhaps the authors can comment on the (expected) frequency of somatic hybridisation in the Ptg population in the discussion of their manuscript.

Reviewers' comments:

Reviewer #1 (Remarks to the Author):

Review of Li et al NComms

This report by Li et al describes the use of haplotype phased genome assemblies for two isolates of the wheat stem rust fungus *Puccinia graminis* to infer a historical hybridization event. The authors use PacBio reads to achieve highly contiguous, phased contigs, and combine this with Hi-C linking reads to infer chromosome organization. Comparing the two assemblies revealed that both share one haplotype at high identity, and confirm that this consists of chromosomes from a single nucleus with one exception. The second haplotype shows much lower sequence similarity to the first and between the two isolates.

Overall, the authors present an exciting story of hybridization in Pgt supported by state of the art genomic data and approaches. My main comment would be a concern that too much has been distilled down to examples and diagrams, with the bulk of the data analysis not depicted in the main text figures and tables. While this journal has a general audience, I found this to be a difficult read in places in needing to jump back and forth to understand what was done. However, these are minor issues in presenting the work behind an important piece of genomic detective work for this field. A larger question for future work is how this hybridization event may have led to the emergence of Ug99 in particular as a highly virulent lineage.

We would like to thank this reviewer for his/her positive comments. We have revised the text to try making it easier for readers to follow the story. Unfortunately, the detailed nature of much of the data analysis requires presentation in supplemental materials.

Minor comments:

Line 71: The call out here for Supplementary Table 1 is unclear as there is no mention of virulence pathotypes. Please add the name/number of this supplementary table to the header of the file (also Table S5), as when downloading in bulk it is not clear which table this is.

We included a header for both supplementary tables and revised the text to refer to supplementary table 1 which confirms the virulence profiles of the isolates in the Ug99 lineage. Please see lines 84-86.

Line 72: In describing the assembly, add the contig counts and N50 to give better context to the synteny mapping that follows.

We have added these details in line 89-91.

Line 75: Are the "conserved fungal genes" the BUSCO set shown in Figure 3d?

No, this is the BUSCO set in supplementary table 4. To convey this message we modified the text in lines 94-95.

Line 77: Explain the equation $n=123$

This was intended to read n=18 and 23 the reference. We have amended the text to read “the known haploid chromosome number of eighteen²³” (line 97).

Line 77-80: How do the bins compare to chromosome number?

As noted the chromosome number is 18. We have referred to this in the text (lines 97, 102)

Figure 2- I found this figure difficult to follow, to understand where the insertions are relative to the gene models. It would help to only have a single frame of reference for depicting the gene structures- the arrows being a different size than the track with coordinates complicated this image. Does AvrSr50 include several exons, or is that representing an insertion in that region?

We have revised the figure and think it is easier to interpret. Exons are not indicated in the figure to reduce complexity, only gene bodies. We also updated the figure legend (lines 875-879)

Line 98: Here the authors note very low identity between the haplotypes; why are these values so much lower than the identity reported at line 135? From Supplementary Table 6 it appears this is due to considering non-aligning regions in calculating identity- was that also the case for the calculations at line 135?

One of the values in figure 3b was found incorrect (it should have been 68% rather than 61%), but nevertheless the identity between haplotypes of this chromosome is lower than for the rest of the genome, which seems to reflect a higher proportion of large structural variants. Yes, the calculations here and in line 135 consider non-aligning regions in calculating overall sequence identity. The text was amended in lines 125-127 to make this clear. We corrected Figure 2b, no changes were made to the figure legend.

Figure 3a and b- Can the authors show plots of all the subtracted read alignment data to illustrate how the coverage thresholds were used to select the haplotypes? These example diagrams while clear provide idealized illustrations of the approach.

While figure 3a and b showed examples of read depth for four contigs, the full data for all contigs is present in supplementary table 7. As suggested we have also now included plots of the subtracted read ratios in Supplementary figure 3 a,b. These plots illustrate the distribution of read coverage for the different classes on contigs to help visualize the cutoffs set for the karyon assignments (as described in lines 434-439). We revised the legend for Supplementary Figure 3.

Line 111- the text on conserved fungal genes refers to Figure 3c; this data is showing in Figure 3d.

We have corrected this in the text (line 146)

Line 112- the text states that “the haplotypes were highly contiguous” referring to 3d, which is the BUSCO data.

We corrected this to refer to figure 4 (line 147)

Line 112-5- For the text on alignment identity here, the methods describe calculating identity from mummer whole genome alignments; this would not include a correction for regions that did not align as used for the Avr regions?

In fact, these numbers did include corrections for non-aligned regions. We have modified the text to include proportion aligned and both overall identity and identity of aligned regions (line 150-152)

Figure 5- In the context of this model, is there supporting data on the level of recombination in Pgt during the sexual stage of its lifecycle? Has LD been estimated previously; this could also be done from the population genomic data.

Unfortunately, LD has not been calculated for Pgt. We do not think that the population used here would be suited for this as we have so many apparent somatic hybrids and we do not know if any of the isolates actually differ by sexual recombination. The level of sexual recombination in the flax rust fungus, which also has 18 chromosomes, has been measured and it is ~115 events/haploid genome (Anderson et al., 2016, BCM Genomics 17, 667) we have added this to the text (lines 170-172)

Figure 6- In the figure legend, the descriptions of panels d and e are swapped.

Figure 6 has been separated into figure 6 and 7 and the legend has been corrected (lines 904-917).

L144: What significance test was used to evaluate Hi-C read mapping in Table S10?

We used a *Chi* square test to assess the deviation of each read pair distribution from a 1:1 ratio as a null hypothesis. Table S10 was expanded to include the p values as well as data for a genome wide comparison.

L168: The statement about “no recombination events” may need some clarification. Were no SVs detected between the two A genomes? There are some off diagonal alignment segments show in in Figure 4a- was there a size threshold (ie the 5 gene block used for haplotype analysis) to detecting rearrangements, such that small events may not have been counted?

The off-diagonal lines in dot plot in figure 4a result from inefficiency of the sorting algorithm in D-genies when so many contigs are used in each dataset and do not represent re-arrangements between contigs. We have included an additional dot plot in fig. 6c using the chromosome assemblies of Pgt21-0 and aligning those to the Ug99 A contigs, which shows no rearrangements. Text included to convey this message is in lines 204-205.

According to the Assemblytics output, there were only 4 structural variants larger than 10 kbp between haplotype A in Ug99 and haplotype A of Pgt21-0. These included 3 deletions and 1 tandem duplication in Ug99, so they did not represent novel genetic content in the isolate. We modified the text to include this information (lines 192-195).

L177: It appears that the ortholog mapping also supports the translocations described at line 154.

Yes, this is true and we have included this in the text lines 230-231.

The section on phylogenetic analysis starting on line 183 refers to Figure 7, however this data is in figure 8. There also appears to be a discrepancy in the figure legend. The text notes a very similar clade of isolates using the A genome referring to Fig 7b (line 197). In Figure 8, the legend stated that panel 8b has the A genome, however this does not contain a very similar clade- that data appears to be in panel 8c.

Yes, this was a mistake and we have corrected the figure citations throughout the manuscript.

The sequence data, both the raw data and the annotated assemblies, needs to be made available in NCBI. PRJNA516922 currently returns no results.

Yes, all data will become downloadable when the manuscript is accepted. NCBI is currently finalising the formatting of the assemblies for release. These will also be accessible through the JGI Mycocosm site.

Reviewer #2 (Remarks to the Author):

In this well-written manuscript, the authors provide clear and convincing evidence for a role of parasexuality and somatic hybridisation in dikaryotic Basidiomycete fungi. Although well-known from haploid, monokaryotic, Ascomycete fungi, this process has not been shown to operate in Basidiomycetes. With their findings, the authors convincingly demonstrate how the Ug99 lineage of the wheat stem rust pathogen, *Puccinia graminis* f. sp. *tritici* evolved. The approach that was used by the authors is sound, state-of-the-art and elegant, and combines genome sequencing with HiC scaffolding, resulting in haplotype phased genome assemblies. Their conclusions are well-supported by the data.

We thank the reviewer for such positive comments about our work.

I appreciate the work that the authors have performed and enthusiastically support publication of this manuscript. My only, minor, concern would be to suggest the authors to perhaps re-consider their final statement in the abstract where they state: "Generation of genetic variation by nuclear exchange may favour the evolution of dikaryotism by providing an advantage over diploidy". Although carefully phrased, the authors seem to suggest that one or the other, dikaryotism or diploidy, may be better or "preferred", while I do not think that evolution necessarily works towards the domination of one system. Clearly, sexual reproduction has advantages, but also comes at a cost. Throughout evolution, several mechanisms evolved to compensate for the lack of advantages of sexual reproduction in case asexual reproduction became the norm. As such, generation of genetic variation by nuclear exchange may just be a means to compensate for loss of sex, although it needs to be noted that plenty Ptg lineages are sexually active, in case the barberry host is available.

We appreciate the reviewer's point and have modified the abstract as suggested (lines 34-36).

Finally, perhaps the authors can comment on the (expected) frequency of somatic hybridisation in the Ptg population in the discussion of their manuscript.

It is difficult to answer this question since we do not have an extensive collection of isolates. Based on our analysis there has been likely 4 somatic hybridisation events involving the three haplotypes defined so far. While this may suggest that these events are relatively rare in time, considering that we are spanning 100 years of race 21 clonal lineage, in the absence of frequent sexual recombination they have made a significant contribution to diversity. We have noted this in the text (line 295-296).